# Phoneme-Aware Hierarchical Augmentation and Semantic-Aware SpecAugment for Low-Resource Cantonese Speech Recognition

**DOI:** 10.3390/s25144288

**Published:** 2025-07-09

**Authors:** Lusheng Zhang, Shie Wu, Zhongxun Wang

**Affiliations:** 1School of Physics and Electronic Information, Yantai University, Yantai 264005, China; ytdxeduzls@s.ytu.edu.cn (L.Z.); wushie@ytu.edu.cn (S.W.); 2Shandong Data Open Innovation Application Laboratory of Smart Grid Advanced Technology, Yantai University, Yantai 264005, China

**Keywords:** low-resource speech recognition, phoneme-level augmentation, specaugment, cantonese asr

## Abstract

Cantonese Automatic Speech Recognition (ASR) is hindered by tonal complexity, acoustic diversity, and a lack of labelled data. This study proposes a phoneme-aware hierarchical augmentation framework that enhances performance without additional annotation. A Phoneme Substitution Matrix (PSM), built from Montreal Forced Aligner alignments and Tacotron-2 synthesis, injects adversarial phoneme variants into both transcripts and their aligned audio segments, enlarging pronunciation diversity. Concurrently, a semantic-aware SpecAugment scheme exploits wav2vec 2.0 attention heat maps and keyword boundaries to adaptively mask informative time–frequency regions; a reinforcement-learning controller tunes the masking schedule online, forcing the model to rely on a wider context. On the Common Voice Cantonese 50 h subset, the combined strategy reduces the character error rate (CER) from 26.17% to 16.88% with wav2vec 2.0 and from 38.83% to 23.55% with Zipformer. At 100 h, the CER further drops to 4.27% and 2.32%, yielding relative gains of 32–44%. Ablation studies confirm that phoneme-level and masking components provide complementary benefits. The framework offers a practical, model-independent path toward accurate ASR for Cantonese and other low-resource tonal languages. This paper presents an intelligent sensing-oriented modeling framework for speech signals, which is suitable for deployment on edge or embedded systems to process input from audio sensors (e.g., microphones) and shows promising potential for voice-interactive terminal applications.

## 1. Introduction

With the rapid advancement of information technology, ASR has become an indispensable pillar of human–computer interactions [1]. Powered by deep learning architectures and statistical modeling, ASR systems extract salient features from speech signals and align them with corresponding acoustic models, thereby enabling computers to “understand” human language. This technology now underpins a wide array of applications—including voice assistants, real-time translation, speech dictation, smart-home control, and call-center automation—bringing unprecedented convenience and efficiency to human–machine communication. Most contemporary ASR solutions are deployed on intelligent voice terminals that rely on acoustic sensors (e.g., microphones), such as smart speakers, in-vehicle infotainment systems, and IoT edge devices. Accordingly, improving the robustness of speech-perception models in low-resource scenarios has become a pressing challenge for real-world sensor-level deployments of ASR.

Despite the impressive advances achieved in Automatic Speech Recognition for mainstream languages such as English and Mandarin, this success is still largely predicated on the availability of extensive labelled corpora [2,3]. Even state-of-the-art architectures that combine the strengths of self-attention and convolution—most notably the Conformer-Transducer [4,5]—are ultimately bounded by the scale and quality of their training data. When the amount of annotated speech drops below the critical threshold of roughly 200 h, a model’s ability to generalize to phonemic variations and dialectal accents degrades sharply [6]. End-to-end neural ASR already attains near-human accuracy (word-error rate <5%) for high-resource languages, yet its performance plummets in low-resource settings. Although self-supervised pre-training (e.g., HuBERT) [7] and multilingual transfer techniques (e.g., mSLAM) [8] can partially compensate for data scarcity, low-resource systems still face persistent challenges—including the pronounced phoneme diversity of regional dialects and the acoustic–semantic mismatch introduced by coarse-grained data augmentation strategies.

Cantonese, a major Sinitic variety spoken chiefly in Guangdong Province, southeastern Guangxi, Hong Kong, and Macao, is renowned for its rich tonal inventory and complex phonological structure, which lend the language great diversity and flexibility in oral expression. Yet, compared with Mandarin, Cantonese suffers from a relative scarcity of labelled speech corpora; its highly colloquial style and strong regional lexicon further exacerbate the difficulty of Automatic Speech Recognition. The challenge is especially acute in low-resource scenarios: with only limited annotated data available, it remains an open—and critical—problem to capture and faithfully reconstruct Cantonese phonetic characteristics for accurate ASR.

In recent years, research on low-resource ASR has converged on three main technical avenues: (i) self-supervised representation learning, (ii) phonology-aware masking augmentation, and (iii) cross-modal dialect adaptation. Self-supervised models such as wav2vec 2.0 [9] and HuBERT [7] have already achieved notable progress for languages like Vietnamese and Thai. Most recently, a joint team from Shanghai Jiao Tong University and Tencent AI Lab introduced VietASR, which combines HuBERT pre-training with a Zipformer architecture to deliver production-grade Vietnamese ASR using only 50 h of labelled speech [10]. Shen et al. probed the internal layers of wav2vec 2.0 and related models, demonstrating that tone categories in tonal languages (e.g., Mandarin and Vietnamese) form linearly separable clusters—thereby motivating specialized modeling for low-resource tonal ASR [11]. In 2023, Li Rui’s group proposed a policy-driven dynamic masking strategy that overcomes the limited diversity of the original fixed-masking approach [12].

However, existing studies show that conventional audio-level augmentation techniques—such as tempo perturbation and additive noise [13]—while increasing data volume, fall short of faithfully modeling the subtle phonemic variations found in low-resource languages like Cantonese. Moreover, the prevailing SpecAugment approach [14,15], which relies on random time–frequency masking, largely overlooks the phonological structure of the speech signal; as a result, critical acoustic cues may be masked indiscriminately. For phoneme-rich and tone-sensitive languages such as Cantonese, this blind masking can blunt a model’s ability to detect phoneme boundaries and semantically salient regions, ultimately degrading recognition performance.

To address the above issues, this paper proposes a unified framework of phoneme-aware hierarchical augmentation and semantics-sensitive masking and systematically evaluates it under low-resource Cantonese conditions. First, a learnable PSM is introduced. Built on a 92 × 92 phoneme confusion matrix generated early in training, the PSM automatically selects easily confusable pairs and, with the aid of Tacotron-2, acoustically resynthesizes the corresponding segments while keeping the original transcription unchanged. During data augmentation, multi-layer attention distributions from wav2vec 2.0 are further combined with BERT-based keyword localization to achieve semantics-sensitive masking. Experiments on the Common Voice Cantonese corpus show that, with 50 h of labelled data, the character error rate (CER) of wav2Vec 2.0 is reduced from 26.17% to 16.88% and that of Zipformer from 38.83% to 23.55%. With 100 h, the CERs drop to 4.27% and 2.32%, respectively, confirming the effectiveness of the proposed method for low-resource Cantonese ASR.

## 2. Basic Theory

Traditional Automatic Speech Recognition (ASR) systems are typically modular, comprising an acoustic model, a language model, and a pronunciation lexicon, among other components. Because these modules are inter-dependent, the overall architecture is complex, and errors from individual modules readily accumulate in the final output. Moreover, conventional systems rely heavily on extensive prior knowledge: crafting a pronunciation dictionary and language model, for example, requires deep expert involvement—an obvious limitation when dealing with dialects or low-resource languages.

In recent years, the rapid progress of deep learning techniques has propelled end-to-end (E2E) ASR models to the forefront of research and industrial deployment [16]. An E2E approach uses a single neural network to map speech directly to text, thereby streamlining the architecture and reducing both intermediate processing stages and dependence on handcrafted priors. As a result, these models exhibit greater robustness and adaptability in low-resource, multilingual, and noisy environments.

### 2.1. Principles of Connectionist Temporal Classification (CTC)

Among various E2E frameworks, CTC [17,18,19] is one of the most widely adopted. CTC introduces a special blank symbol to handle the length mismatch between an input speech frame sequence and its target transcription. At each time step, the network may output either a real character or a blank; this mechanism yields an implicit alignment.

During training, CTC does not require frame-level annotations. Instead, it sums the probabilities of all possible alignments to build its loss and optimizes parameters via back-propagation. Let the input acoustic sequence be x, let the target text sequence be y, and let $\beta^{-1}$ (y) denote the set of all alignment paths that collapse (via the mapping of β to y after removing blanks and repeated symbols. Then, the CTC loss is defined as(1)$$L_{\mathrm{CTC}}=-\ln P(y|x)=-\ln\sum_{\pi \in \beta^{-1}(y)}\prod_{t=1}^{T}p_{\pi_{t}}(t),$$

Here, T denotes the number of frames in the input speech sequence, and $p_{\pi_{t}}(t)$ represents the probability that the model outputs a character or blank symbol at time step t.

During the decoding phase, CTC typically employs greedy search or beam search to identify the most probable output sequence. Since the CTC model does not rely on a predefined lexicon or acoustic priors, its simplified structure and adaptive capability make it highly effective in multilingual, dialectal, and low-resource speech recognition tasks.

### 2.2. The wav2Vec 2.0 Model

wav2Vec 2.0 [9], proposed by the Facebook AI team, is a self-supervised speech pre-training model designed to extract high-quality acoustic representations directly from raw waveforms. During pre-training, the model leverages large amounts of unlabeled audio: by masking portions of the latent features and forcing the network to predict the masked content, it learns rich phonetic patterns and contextual information without costly manual annotation.

wav2Vec 2.0 first employs a stack of convolutional layers to capture local time domain features then feeds the resulting sequence into Transformer blocks to model long-range dependencies, thereby producing higher-level and more generalizable representations. In its self-supervised phase, wav2Vec 2.0 is typically optimized with a contrastive loss, which can be formulated as follows:(2)$$L_{\mathrm{contrast}}=-\log\left(\frac{\exp\left(\frac{\mathrm{sim}(c_{t},q_{t})}{\tau }\right)}{\exp\left(\frac{\mathrm{sim}(c_{t},q_{t})}{\tau }\right)+\sum_{k=1}^{K}\exp\left(\frac{\mathrm{sim}(c_{t},q_{k})}{\tau }\right)}\right),$$
where sim(·,·) denotes the similarity function (e.g., cosine similarity or inner product), τ is a temperature hyper-parameter, k is the number of negative samples, $c_{t}$ denotes the context vector at time step t, $q_{t}$ is the target representation vector at that time step, and $q_{k}$ is a negative sample vector distinct from the current $q_{t}$. Following the original wav2vec 2.0 configuration, we randomly sample k = 100 negative vectors for each masked time step (drawn from other unmasked frames within the same batch). Empirically, this setting keeps GPU memory usage manageable while still providing stable contrastive gradients. After pre-training, wav2Vec 2.0 is fine-tuned on labelled data to perform an end-to-end mapping from speech to text, exhibiting strong robustness in noisy environments, dialectal speech, and multilingual settings. To address low-resource scenarios, the wav2vec2-large-xlsr-53 variant is pre-trained on large-scale audio spanning 53 languages; by sharing cross-lingual acoustic representations, it provides more favorable initialization for low-resource ASR and significantly improves overall recognition performance.

### 2.3. The Zipformer Model

Zipformer is an efficient encoder architecture purpose-built for ASR, built on, and extending, the Conformer architecture [20]. By seamlessly fusing self-attention with convolution and introducing a sparse-attention strategy, Zipformer lowers the Transformer’s computational burden on long sequences while still capturing global context with high fidelity.

Zipformer employs a U-Net-style, multi-resolution framework that alternates downsampling and upsampling to learn time-series representations at multiple frame rates. The model first applies a convolutional embedding layer to the raw 100 Hz acoustic features, reducing the frame rate to 50 Hz and projecting the signal into an initial latent space. The resulting 50 Hz sequence is then processed by six stacked modules that alternate downsampling and upsampling, operating successively at 50 Hz, 25 Hz, 12.5 Hz, 6.25 Hz, 12.5 Hz, and 25 Hz. Each module interleaves self-attention with convolution, enabling the encoder to capture rich contextual information across multiple temporal scales. Finally, the outputs of all modules are merged—after appropriate truncation or zero-padding—and passed through a downsampling stage that standardizes the frame rate at 25 Hz, yielding the encoder’s ultimate feature representation.

Unlike standard multi-head self-attention, Zipformer splits the attention computation into two critical stages: (1) It first forms the attention weight matrix A∈ℝN×D for the input sequence. This step mirrors the conventional soft-max (QKT/dk) computation, except that only the weight matrix A is retained. The formulation is given by(3)A=soft max(QKT/d),Q=HWQ,K=HWk,
where H∈ℝN×D is the feature matrix, N is the number of time steps, and D is the feature dimension. This step mirrors the conventional soft-max over the scaled dot-product but preserves only the weight matrix A. Q and K denote the query and key vectors obtained from linear projections of the input sequence, encoding the pairwise matching relationships across different time steps. (2) Once A is obtained, all subsequent self-attention (SA) blocks and the newly introduced non-linear attention (NLA) blocks reuse this same matrix to perform diverse feature transformations across multiple sub-modules—without recomputing the expensive QKT product.

By injecting sparse attention at the encoding stage, Zipformer lowers computational complexity while retaining the capacity to capture global information. The model also employs BiasNorm in place of conventional LayerNorm, preventing the normalization failure that can occur early in training. Through hierarchical downsampling within the encoder, the systematic reuse of attention weights, and improved normalization and activation functions, Zipformer achieves more efficient modeling of long speech sequences. Alternating across multiple frame rates markedly reduces the cost of large-scale attention, affords a wider receptive field at low frame rates, and preserves finer details at high frame rates—giving the model greater potential to capture both long-term dependencies and short-term dynamics.

As shown in Figure 1, Zipformer employs a U-Net-like multi-resolution architecture that learns temporal representations via a series of downsampling and upsampling stages. The model first applies a single convolutional embedding to the original 100 Hz acoustic features, reducing the frame rate to 50 Hz and projecting into the initial embedding dimension. The resulting 50 Hz feature sequence is then fed into a symmetric stack of six modules, which process at 50 Hz, 25 Hz, 12.5 Hz, 6.25 Hz, 12.5 Hz, and 25 Hz, respectively. Each module consists of downsampling and upsampling operations, integrating self-attention and convolution to capture contextual dependencies across different temporal resolutions. The outputs of these modules are appropriately cropped or zero-padded before being fused and then passed through a downsampling block to unify the frame rate at 25 Hz, yielding the final encoder representations.

### 2.4. Phoneme-Level Augmentation

At the text level, we first introduce phoneme-scale perturbations: the original Cantonese transcription is converted to a phoneme sequence using a pinyin-to-phoneme mapping, and phonemes that are easily confused or closely articulated in Cantonese are stochastically substituted under a constraint of strict semantic preservation. The resulting “pseudo” transcripts therefore retain the original meaning while injecting accent variation and slight pronunciation shifts, furnishing the acoustic model with richer pronunciation variants.

At the speech level, the original audio and its perturbed transcript are force-aligned with MFA to obtain the onset and offset of each phoneme. The waveform is then segmented into phoneme-level chunks, which are locally inserted, replaced, re-ordered, or time-stretched/compressed according to the perturbed sequence; cross-fade windows are applied at splice boundaries to smooth discontinuities. This procedure emulates accent, speech rate, and co-articulation variability while keeping labels unchanged.

Finally, the augmented samples are mixed with the original corpus for training. Experiments demonstrate that this strategy broadens pronunciation diversity and boosts robustness: after adding phoneme-level augmentation, the model’s average CER on a multi-accent test set drops by approximately 9%, clearly outperforming conventional signal- or spectrogram-level augmentation. The method thus offers an efficient, low-cost avenue for optimizing low-resource, end-to-end Cantonese ASR.

### 2.5. Learnable Phoneme Replacement Matrix

This paper uses the model to perform forced decoding on the development set and compares the outputs frame-by-frame with MFA-aligned phonemes. A 92 × 92 count matrix C is then compiled and, after normalization, converted into conditional probabilities. Setting the threshold = 0.03, we retain reciprocal entries satisfying and yielding 248 pairs—5.9% of the theoretical combinations (92 × 92 − 92 = 8372 off-diagonal directed entries). To verify the screening quality, we randomly sampled 100 pairs during the offline analysis stage for a listening check (results not used for training), achieving 93% consistency. For each, this paper re-normalizes by obtaining the replacement probabilities. During training, up to two random replacements are sampled per utterance to preserve semantic content. The original audio is first segmented into phoneme-level fragments using MFA boundaries; for each fragment selected for replacement, Tacotron-2 synthesizes the corresponding target phoneme.

### 2.6. Improved SpecAugment

We propose an adaptive augmentation framework that fuses model-attention weights with textual semantics and employs reinforcement learning to discover optimal perturbation parameters. First, attention maps from a pre-trained wav2vec 2.0 model are extracted for each training utterance to locate time regions the network “focuses on”. By analyzing multi-head attention matrices layer-wise, frames assigned high weights are identified and subjected to stronger perturbations. Early in training, only mild time masking is applied; the masking probability and span are gradually enlarged, and frequency masking plus light Gaussian noise are injected within the same regions to simulate “information loss”. Masking position and length are dynamically modulated by attention intensity—the higher the weight, the larger or more likely the mask—forcing the recognizer to rely on context rather than over-fitting to salient local cues.

Second, we selectively mask audio segments corresponding to key lexical items derived from the transcripts. The text is tokenized and POS-tagged to extract content words that carry critical information; MFA then supplies their time spans in the audio. Time masking these keyword segments compels the model to infer the missing words from surrounding context, implicitly enhancing its language-modeling capacity.

Finally, a reinforcement learning (RL) controller—implemented as an RNN policy network—outputs SpecAugment hyper-parameters (number of masks, duration, time-warp magnitude, etc.) at each training iteration. These parameters are applied to the current mini-batch, and the controller is updated using the validation set error as feedback, gradually converging on an augmentation strategy tailored to the dataset.

Finally, we incorporate a reinforcement learning technique to adaptively optimize the SpecAugment parameters. Conventional SpecAugment requires manual tuning of the number of masks, mask lengths, and time-warp magnitude, yet the optimal settings differ across datasets. Drawing on the neural architecture search (NAS) concept introduced by Zoph and Le (2017) [21], we build an RNN controller as the policy network. At each training iteration, the controller outputs a set of SpecAugment parameters, applies them to the current mini-batch, and updates itself using the model’s validation error as feedback. Through repeated exploration and policy updates, it gradually converges on an optimal augmentation strategy. The policy network is a single-layer LSTM with 128 hidden units that produces a 360-dimensional soft-max distribution and is updated according to the REINFORCE rule:(4)$$L_{\mathrm{ctrl}}=-(R_{t}-b)\log\pi_{\theta }(a_{t}|s_{t}),$$

Here, b is a moving-average baseline used to reduce gradient variance, $a_{t}$ denotes the action sampled by the controller at step t, and $s_{t}$ represents the current state. The reward function is defined as follows:(5)$$R_{t}=-DevCER_{t}-0.1{\left\|\lambda_{t}-0.2\right\|}_{1},$$

In other words, the primary reward is the negative CER on the development set, with an additional penalty for excessive masking (empirically, a mask probability of 0.2 tends to cause underfitting).

## 3. Training Procedure

### 3.1. Dataset and Pre-Processing

We conduct all experiments on the Cantonese subset of the Common Voice corpus, which contains volunteer recordings from Hong Kong, Guangdong, and neighboring regions. This subset spans a wide range of accents, genders, ages, and speaking styles and includes informal, conversational utterances that closely mirror real-world usage, making it ideal for evaluating practical ASR performance.

All audio is resampled to 16 kHz. The original train/dev/test split ratios are preserved. For each utterance we compute 80-dimensional log-Mel filter-bank features, followed by per-frame mean–variance normalization. The reference transcription is converted into a sequence of token IDs (including the CTC blank), and superfluous metadata fields are removed.

To satisfy the CTC requirement of equal-length inputs within a batch, we employ dynamic padding: a custom collator pads each batch to the length of its longest sequence and produces the corresponding attention masks, ensuring computational efficiency and consistent tensor shapes.

### 3.2. Model Training

We initialize from the multilingual wav2vec 2.0 checkpoint and fine-tune it for Cantonese ASR. The acoustic encoder comprises 7 temporal convolution layers; the context network is a 12-layer Transformer with 12 attention heads per layer and a 64-dimensional head size.

Optimization is performed with Adam [22] using a linear warm-up followed by cosine annealing of the learning rate. Each GPU processes a mini-batch of 32 utterances, and training is accelerated with FP16 mixed precision to reduce memory consumption.

To curb over-fitting, we adopt early stopping: if the validation loss fails to improve for five consecutive evaluations, training halts. For comparability across experiments, we cap training at 30 epochs, but early stopping never interrupts the first full pass through the data. After every epoch, we evaluate on the development set and save the best checkpoint for subsequent testing and analysis.

### 3.3. Loss Functions

wav2Vec 2.0 is a self-supervised speech representation framework whose core idea is to encode raw audio signals, mask portions of the encoded sequence, and then learn useful speech features through contrastive learning. During training, the model randomly masks some time-step features, uses a Transformer to extract contextual information, and is trained by matching the contextual representations to the discretized true target representations. Its overall loss function consists of two parts: the contrastive loss and the diversity loss.

Contrastive loss: For each masked time step t, the model generates a context representation vector $q_{t}$; meanwhile, the discretization module yields the true target representation vector $k_{t}$ (the positive sample). To enable the model to distinguish the correct target from negative samples (incorrect representations sampled from other time steps), the contrastive loss is introduced and is defined as follows:(6)$$L_{\mathrm{contrastive}}=-\sum_{t\in M}\log\frac{\exp\left(\mathrm{sim}(q_{t},k_{t})/\kappa \right)}{\exp\left(\mathrm{sim}(q_{t},k_{t})/\kappa \right)+\sum_{k^{-}\in N_{t}}\exp\left(\mathrm{sim}(q_{t},k^{-})/\kappa \right)},$$

Here, M denotes the set of all masked time steps, $\mathrm{sim}(q_{t},k_{t})=\frac{q_{t}^{⊤}k_{t}}{\left\|q_{t}\|\cdot \|k_{t}\right\|}$ is the similarity function, k is a temperature parameter that controls the smoothness of the distribution, and 𝒩_t_ is the set of negative samples drawn for time step t. This loss encourages the model to distinguish the correct target representation from negative samples during prediction, thereby learning more discriminative features.

Diversity loss: To prevent the quantization module from collapsing, wav2Vec 2.0 also adds a diversity loss. This term encourages the model to use all discrete codes in the codebook as evenly as possible. Its formal definition is as follows:(7)$$L_{\mathrm{diversity}}=-\sum_{j=1}^{K}\frac{1}{K}\log p_{j},$$

Here, K denotes the size of the codebook, and $p_{j}$ represents the probability that the j-th code entry is selected over the course of training. In this way, the model is encouraged to give every code entry a chance to be used, thereby increasing the diversity and utilization of the discrete representations.

Total loss: Ultimately, wav2Vec 2.0 is trained by jointly minimizing the contrastive loss and the diversity loss; the overall loss function is therefore given by(8)$$L=L_{\mathrm{contrastive}}+\lambda L_{\mathrm{diversity}},$$

Here, *λ* is a hyper-parameter that balances the weights of the two loss components. This combined strategy not only ensures that the model learns highly discriminative features during self-supervised training but also guarantees effective utilization of the codebook in the quantization module, thereby leading to improved performance on downstream tasks.

### 3.4. Semantics-Sensitive Masking

This paper introduces semantics-sensitive SpecAugment (SAM). SAM first leverages the multi-layer attention weights of wav2vec 2.0 to extract temporally critical frames, then combines BERT-based content-word localization with MFA alignments to delineate semantic regions. A single multi-dimensional masking map is generated in one shot—replacing the time- and frequency-masking hyper-parameters that conventional SpecAugment must tune via grid search—thereby enabling finer-grained and more robust data augmentation.
(1)Attention masking: This paper extracts the frame-level average attention $\alpha_{t}$ from layers 6 to 9 of wav2vec 2.0, normalizes it across all frames, and computes the Bernoulli selection probability, as computed by Equation (9):(9)$$p_{t}=\lambda_{t}\frac{\alpha_{t}}{\alpha },\;\alpha =\frac{1}{T}\sum_{t=1}^{T}\alpha_{t},$$Here, $\lambda_{t}\in \left(0,1\right)$ denotes the baseline proportion for temporal masking.(2)Keyword masking: For the transcription text $T=\{w_{1},\ldots ,w_{N}\}$, this paper applies jieba-POS for word segmentation and retains only content words, yielding n candidate $\mathrm{token}{\left\{w_{i}\right\}}_{i=1}^{n}$. Let $w_{i}$ denote the i-th subword unit, whose character offset in the original sentence is $\left[p_{i}^{st},p_{i}^{ed}\right]$. These tokens are then passed through BERT to obtain their embedding vectors $h_{i}$. Finally, a learnable gating vector is used to compute the semantic salience of each token, as computed by Equation (10):(10)$$s_{i}=\frac{\exp\left(u^{⊤}h_{i}\right)}{\sum_{j}\exp\left(u^{⊤}h_{j}\right)},$$
and Top−k=2 is set to define the semantic focus set.(3)Joint masking and spectrogram update: The e of each token is computed by Equation (11):(11)$$M(t)=\max\left\{M_{\mathrm{att}}(t),\;M_{\mathrm{sem}}(t)\right\},S_{f,t}^{\prime \prime }=\left\{\begin{array}{ll} v_{\mathrm{mask}}, & M(t)=1\\ S_{f,t}, & M(t)=0 \end{array}\right.$$Here, the uniform fill value is set to $v_{\mathrm{mask}}$ = −80 dB, $S_{f,t}$ denotes the power of the f-th Mel band at frame t, and $M_{\mathrm{att}}(t)$ is the binary indicator function for attention-based masking and represents the Mel spectrogram after both attention and semantic masking. If $\sum_{t}M_{\mathrm{sem}}(t)>0.3T$, the semantically masked frames are randomly downsampled to 0.3 T to avoid excessive information loss.

## 4. Results

### 4.1. Experimental Setup and Evaluation Metrics

All experiments are conducted on the Common Voice corpus. The original audio files are uniformly resampled to 16 kHz, mono channel, and the dataset is split into training, development, and test sets according to the original proportions. Since speaker IDs in the Common Voice Cantonese subset are not publicly available, we were unable to enforce a strictly speaker-independent split. Instead, we partitioned the data into training, development, and test sets according to the official 70%/10%/20% proportions, which may allow speaker overlap between partitions.

In speech recognition research, the most commonly used metric is the error rate. Depending on the modeling unit, typical measures include the phoneme error rate (PER), CER, and sentence error rate (SER). Given Cantonese’s rich inventory of Chinese characters and syllabic representations—and because our recognizer operates at a character-like granularity—the CER is the most appropriate primary metric.

The CER is computed by comparing the recognition output with the reference transcription via the edit distance, which counts substitutions (S), deletions (D), and insertions (I). Its formula is(12)CER=(S+D+I)/N
where N denotes the total number of characters in the reference transcript. This metric directly reflects the discrepancy between the recognition output and the reference text, making it essential for evaluating the model’s performance at the character level.

### 4.2. Spectrogram Analysis

Below, Figure 2a presents the Mel spectrogram of the raw audio, prior to any augmentation. While the spectral features are clearly visible, the absence of augmentation leaves the model less adaptable to acoustic variations.

Below, Figure 2b presents the spectrogram after applying SpecAugment. In the 80-dimensional Mel bands, there are vertical black stripes—temporal masks approximately 4–6 frames long (≈35 ms)—at 11 locations (0.5 s, 0.7 s, 1.3 s, 2.8 s, etc.), which disrupt the continuity of formant trajectories over time. These masked regions simulate missing segments in diverse audio conditions. By occluding portions of the spectrum, the model is compelled to leverage broader contextual information, thereby improving its recognition robustness to new environments, different accents, and even simulating packet loss or limited bandwidth scenarios.

Below, Figure 2c illustrates the Mel spectrogram after phoneme-level augmentation and concatenation on top of SpecAugment. Unlike the previous two, this spectrogram incorporates phoneme-level edits: random time intervals are fully masked (vertical black bars), and the discontinuities in energy/formants at 3.5 s and 4.3 s result from excising and reassembling phoneme segments. This combined augmentation simulates variations in the speaking rate and accent misalignment, enabling the model to accommodate a wider range of speech characteristics and deliver higher accuracy in practical, low-resource ASR applications.

### 4.3. Recognition Performance at Different Data Scales

Table 1 reports the ASR results on the Common Voice 50 h and 100 h subsets, with the CER evaluated on the standard test set.

Table 1 shows the ASR results on the Common Voice Cantonese dataset using 50 h and 100 h of training data, with the CER evaluated on the standard test set. As the amount of training data increases, both models exhibit marked improvements in recognition performance. Specifically, when wav2Vec2’s training data grows from 50 h to 100 h, its CER falls from 26.17% to 6.29%; Zipformer’s CER likewise plunges from 38.83% to 4.16%. Although the two models begin with different error rates under the same conditions, both display a strong sensitivity to the data scale, with recognition errors declining exponentially as more data is incorporated.

### 4.4. Combined Effect of Phoneme-Level Augmentation and Semantic-Aware SpecAugment

Table 2 presents the character error rates obtained after simultaneously applying phoneme-level adversarial augmentation and SAM. Relative to Table 1, the CER of wav2vec 2.0 (50 h) drops from 26.17% to 16.88% (a 35.5% relative reduction), while Zipformer (50 h) falls from 38.83% to 23.55% (a 39.4% reduction). On the 100 h corpus, wav2vec 2.0 and Zipformer further decrease to 4.27% and 2.32%, corresponding to additional gains of 32.1% and 44.2% over their respective baselines.

Phoneme-level augmentation explicitly enriches tonal and homophonic variants, alleviating Cantonese pronunciation sparsity; SAM compels the encoder to infer missing information from context, boosting robustness and language modeling capability. Together, the two techniques complement each other, producing far larger gains than either alone.

### 4.5. Ablation Study: Effect of Improved SpecAugment Only

To measure the contribution of each component, we removed the phoneme-level augmentation and retained only SAM, retraining the models to obtain Table 3.

Although relative to the baseline, the CER of wav2vec 2.0 (50 h) drops from 26.17% to 21.69% and that of Zipformer (50 h) falls from 38.83% to 27.02%, these gains still lag behind the joint approach. SAM alone enhances robustness to missing key information, while phoneme-level augmentation supplies irreplaceable pronunciation diversity; only by combining the two can we achieve the largest and most consistent error reductions across different architectures and data scales.

### 4.6. Error Analysis

On 4884 test utterances, using syllable-level alignment statistics, the PSM + SAM model trained on 50 h achieved an overall CER of 17.5% (95% confidence interval ± 0.4 percentage points). The tone CERs for the six Cantonese tones were (40.8, 47.7, 51.3, 43.3, 47.8, 56.1)%, averaging 47.9%—substantially higher than the overall CER measured without tone markers—indicating that vowels and their associated tone markers remain the primary sources of error. By phoneme category, the vowel CER reached 62.0%, whereas initial consonant and entering tone coda mismatches accounted for less than 1%. Taken together, explicit tone modeling and vowel discrimination remain the key bottlenecks; future work will focus on strategies such as tone embeddings and tone-aware language models to further reduce the tone CER and enhance system robustness.

## 5. Limitations and Prospects

### 5.1. Study Limitations

Although this work has made encouraging progress, several limitations remain to be addressed.

First, the training and validation data are drawn exclusively from the Common Voice corpus, whose coverage is narrow and does not include telephone speech, street-scene noise, or multi-speaker conversations; hence, the model’s out-of-domain robustness has yet to be verified.

Second, while six Cantonese tones are explicitly encoded in the output, the tone character error rate (tone CER) is still 48%, with most confusions occurring between adjacent tones sharing the same rime. This indicates that the model has not fully captured fine-grained F0 distinctions and tone-contour dynamics.

Finally, the perceptual quality and alignment accuracy of the Tacotron-2 replacements have not been quantified; future work will incorporate standard objective metrics (PESQ and STOI) and boundary error analysis to ensure that synthetic artefacts do not impair recognition.

### 5.2. Future Work

Future research will proceed along three directions.

Tone-aware integration: We will embed explicit tone vectors and apply tone-aware language model rescoring within the current framework, allowing learnable tone embeddings to cooperate with semantic information in order to lower the tone CER.

Broader model coverage: Keeping PSM and SAM unchanged, we will introduce additional self-supervised encoders at inference, compare the gains and ceilings of different backbones, and assess portability to other tonal languages such as Minnan, Hakka, Thai, and Vietnamese.

Dynamic hyper-parameter tuning: We will explore a reinforcement learning controller to adaptively optimize key hyper-parameters—including the phoneme substitution threshold and masking ratio—and, through model distillation and INT8 quantization, enable low-latency streaming ASR decoding on edge devices.

## Figures and Tables

**Figure 1 sensors-25-04288-f001:**
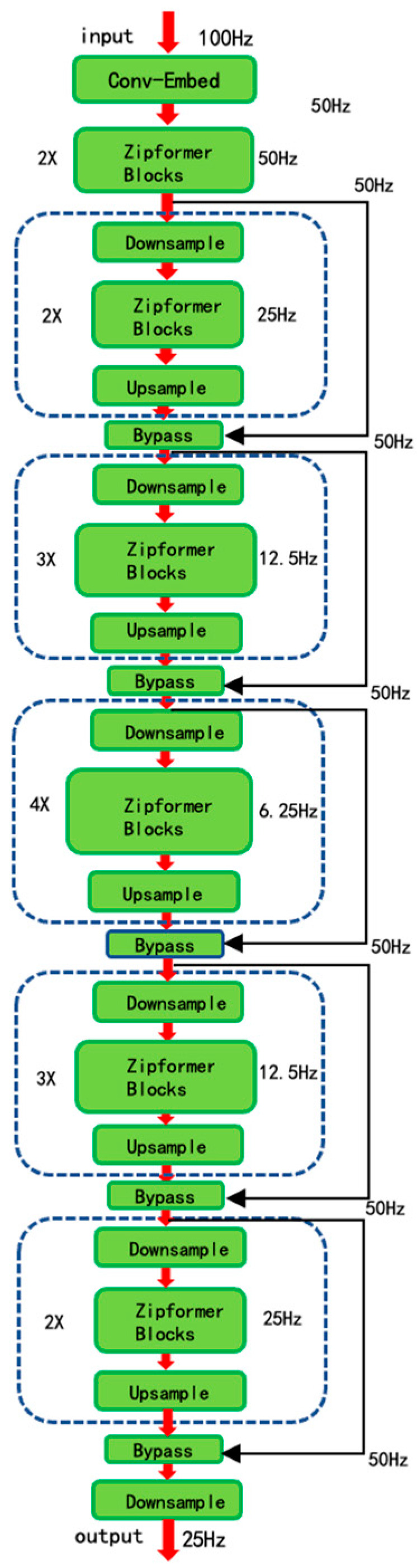
Zipformer architecture diagram.

**Figure 2 sensors-25-04288-f002:**
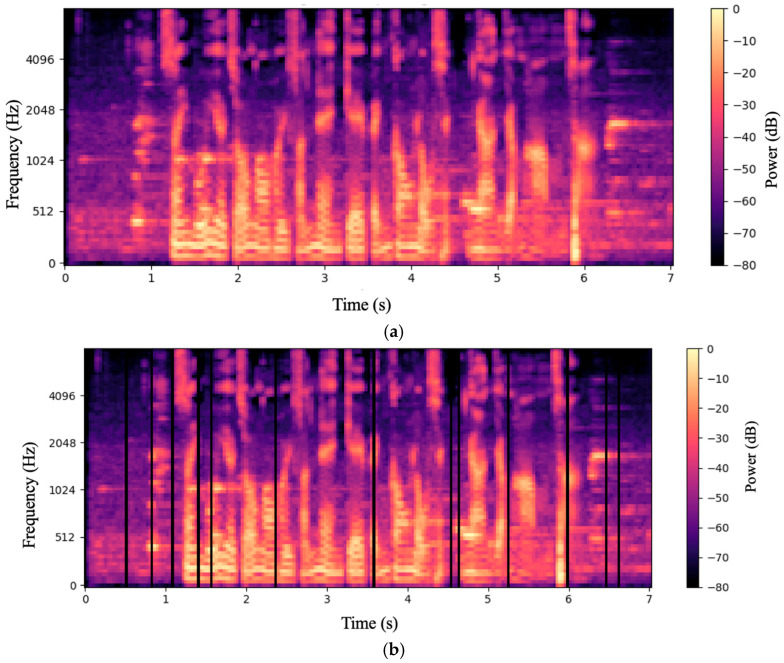
(**a**) Mel spectrogram of the original audio; (**b**) semantics-guided SpecAugment (original audio); (**c**) semantics-guided SpecAugment (spliced audio).

**Table 1 sensors-25-04288-t001:** Comparison of model recognition performance across different training data sizes.

Model	Data Size	CER
wav2vec 2	50 h	26.17
Zipformer	50 h	38.83
wav2vec 2	100 h	6.29
Zipformer	100 h	4.16

**Table 2 sensors-25-04288-t002:** Recognition performance under joint augmentation strategies.

Model	Data Size	CER
wav2Vec2	50 h	16.88
Zipformer	50 h	23.55
wav2Vec2	100 h	4.27
Zipformer	100 h	2.32

**Table 3 sensors-25-04288-t003:** Character error rates of the two models when only SpecAugment augmentation is applied.

Model	Data Size	CER
wav2Vec2	50 h	21.69
Zipformer	50 h	27.02
wav2Vec2	100 h	5.71
Zipformer	100 h	3.53

## Data Availability

The data presented in the study are available at https://commonvoice.mozilla.org/.

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
