# Peer review of "Phoneme-Aware Hierarchical Augmentation and Semantic-Aware SpecAugment for Low-Resource Cantonese Speech Recognition"

_sensors, 2025, doi:10.3390/s25144288_

Round 1
Reviewer 1 Report
Comments and Suggestions for Authors
Contributions:
This study proposes a phoneme-aware hierarchical augmentation framework that enhances performance without additional annotation for Cantonese ASR. My comments are as follows:
- The methods are described as superficial. It is not adequate.
- The proposed method was not compared with state-of-the-art methods. It is not convinced that the proposed method is good.
- (Line 133 on page 4) How do you obtain the number of negative sample K?
- (Page 4) The Zpformer in sub-section 2.3 is unclear. Please provide the network structure and the hyper-parameters.
- (Page 5) The description of Spec-Augment is unclear. Please provide pseudo code to improve the clarity.
- (Line 206 on page 5) How can you selectively mask the audio segment? Please provide equations to express the idea.
- (Lines 211 to 215 on page 5) The detailed procedure for reinforcement learning is too superficial.
- (Lines 253 to 255 on page 6) Vectors should present the contextual representation q and target representation k.
- (Page 6) An equation in eq should present the operator sim. (3).
- (Page 7) Section 3.4 should be moved to section 4.
- (Page 7) The experimental arrangement should be introduced in section 4 first.
- (Pages 7 and 8) Figures 1-3 should be merged.
- (Page 8) The caption and table should be on the same page.
- (Page 9) The position of Table 2 is inadequate.
The quality of the English language should be improved.
Reviewer 2 Report
Comments and Suggestions for Authors
Although the article claims to address the issue of limited resources in Cantonese automatic speech recognition (ASR) through phoneme-aware augmentation and semantic-aware SpecAugment, the proposed techniques merely represent incremental modifications of existing data augmentation and masking strategies, rather than offering any genuinely novel contributions. Similar phoneme- and attention-level augmentation methods have been widely published at ICASSP, INTERSPEECH and NeurIPS in recent years. The absence of comparative methodology within this contemporary literature undermines the justification for the study's relevance.
A significant limitation is the absence of a structured, up-to-date literature review on the latest acoustic-only automatic speech recognition (ASR) solutions for tonal languages. While the introduction briefly references early works, it overlooks recent advancements in self-supervised learning and adaptive augmentation frameworks applied to Cantonese and other low-resource dialects.
The description of the phoneme substitution matrix (PSM) lacks technical rigor. The article does not explain the criteria for selecting confusable phoneme pairs, nor does it address the statistical properties of the substitution matrix or the improvement in phonetic coverage that it introduces. Without quantitative validation or justification based on a confusion matrix, the impact of the PSM remains speculative and insufficiently supported.
Although Tacotron 2 is presented as a tool for synthesising phoneme variants, the article fails to discuss the quality of the generated audio, alignment error rates or how synthetic data artefacts influence recognition accuracy.
The article's description of the reinforcement learning (RL) controller for SpecAugment hyperparameter tuning is inadequate. The authors fail to specify the architecture of the RL agent, its state space, the definitions of its actions, the design of its reward function and its convergence criteria. Consequently, it is unclear whether the RL module meaningfully contributes to performance improvements or merely adds unnecessary complexity.
Another methodological limitation is the fact that the study relied exclusively on the Common Voice Cantonese dataset.
Another issue is the narrow scope of the model benchmarking. While more competitive and recently published self-supervised ASR frameworks are omitted, only Wav2Vec 2.0 and Zipformer are evaluated.
The article does not provide any ablation or error analysis relating specifically to tonal classes, phoneme types or common Cantonese misrecognition patterns.
The figures and tables in the manuscript are overly simplistic. The spectrogram visuals have illustrative value only and lack quantitative metrics, such as changes in the signal-to-noise ratio or phoneme boundary distortions. The tables lack confidence intervals or significance annotations, which limits the interpretability of the reported improvements.
The article omits a dedicated limitations section, which is a standard requirement in rigorous experimental ASR research. Key risks, such as overfitting to small corpora, errors in phoneme alignment, artifacts resulting from augmentation, and model instability under low-resource conditions, remain unacknowledged.
The article exclusively focuses on augmentations at the phoneme and spectrogram levels, yet it disregards recognized complementary techniques for improving the robustness of automatic speech recognition (ASR) in tonal languages, such as pitch contour normalization, tone embedding models, and tone-specific language model rescoring.
While the authors claim that their method can be applied to other tonal languages, this assertion lacks supporting evidence or reasoning. No data or discussion is provided on the phonetic structures of other tonal languages, nor on how PSM and SAM would transfer to them.
Furthermore, the article lacks a concrete plan for integrating phoneme- and semantic-aware augmentation with modern multilingual ASR frameworks, which have already demonstrated superior cross-lingual performance for low-resource languages. Failing to address this issue could render the presented method obsolete in the context of large-scale multilingual models.
Reviewer 3 Report
Comments and Suggestions for Authors
The way in which BERT outputs are mapped to time segments via MFA is unclear. The use of BERT for text semantics is also unclear. The text 'Text-semantic foci extracted by BERT' claims that BERT provides semantic focus. In the text 'Text is tokenised and POS-tagged to extract content words', we see that POS tags are used for keyword masking.
The structure of the RL controller is not detailed. Its input features, reward function and training process are not specified.
Lines 157–158. The text states that H is the feature matrix, but does not clearly explain how A is computed from H.
Line 163: 'QK' is used without defining 'Q' and 'K'.
Figures 1–3 are described minimally.
Standard SpecAugment should be included as a comparison baseline.
There are many uncertainties in the formulas. For example:
In Eq. 1: The text introduces the input as the capital letter 'X', but the formula uses the lowercase letter 'x'. There is an unnecessary space after P in “P (y|x)” and the asterisks for the symbol “t” in “time step *t*” are also unnecessary.
In Eq. 2: The numerator uses ql, which is not defined in the text. The denominator uses both ql and qk, but qk is also undefined. The text uses 'K' in regular font, but variables should be italicised in maths mode.
The same applies to other formulas. All formulas must be verified. All variables must be explained. Their use must be consistent.
Term ““Augmented SpecAugment” is unclear.
Figures have both a title inside the figure and a caption below with the same wording. The in‐figure title should be removed.
Full axis labels i.e., “Time (s)” and “Frequency (Hz)” should be given in figures.
Round 2
Reviewer 1 Report
Comments and Suggestions for Authors
The authors have improved the quality of this paper. It can be accepted after minor revision.
Minor comments:
- The performance in Table 1 should be discussed explicitly.
- (Page 12) The caption of Table 1 is incorrect.
- (Page 12) The caption of Table 2 should be revised.
The quality of the English language should be further improved.
Author Response
Comments 1: The performance in Table 1 should be discussed explicitly.
Response 1:Table 1 shows the ASR results on the Common Voice Cantonese dataset using 50 hours and 100 hours of training data, with CER evaluated on the standard test set. As the amount of training data increases, both models exhibit marked improvements in recognition performance. Specifically, when wav2Vec2’s training data grows from 50 hours to 100 hours, its CER falls from 26.17% to 6.29%; Zipformer’s CER likewise plunges from 38.83% to 4.16%. Although the two models begin with different error rates under the same conditions, both display a strong sensitivity to data scale, with recognition errors declining exponentially as more data is incorporated.
Comments 2: (Page 12) The caption of Table 1 is incorrect.
Response 2:Thank you for catching this. On page 12 I have updated the caption of Table 1 to:Model recognition performance under different data scales.
Comments 3: (Page 12) The caption of Table 2 should be revised.
Response3: Recognition Performance Under Joint Augmentation Strategies.
Comments 4: The quality of the English language should be further improved.
Response4: Thank you for this suggestion. I have thoroughly reviewed and revised the manuscript for English language quality and clarity. All sections have been carefully proofread to improve grammar, style, and readability.
Reviewer 2 Report
Comments and Suggestions for Authors
The authors have significantly revised the article, many points have been clarified, and the first part of the article has begun to read better. In this form, the article may be recommended as it may be useful for the scientific community.
Author Response
Thank you very much for your encouraging words and for taking the time to read through our revised manuscript. We are delighted to hear that the changes we implemented have clarified many points and improved the readability of the first part of the paper. We sincerely hope that our work will prove useful to the scientific community, and we appreciate your recommendation.
Reviewer 3 Report
Comments and Suggestions for Authors
The equations have mixed numbering (some reuse the same numbers multiple times).
The test data should be speaker-independent if claiming generalization. This is not clarified.
“wav2vec 2.0” vs “Wav2Vec 2.0” (Inconsistent capitalization)
Uses mixed % and percent (unify style)
Comments on the Quality of English Language
Phoneme Error Rat (should be Rate)
coarse-grained data-augmentation strategies (should be coarse-grained data augmentation strategies)
This paper introduce Semantics-Sensitive ... (should be This paper introduces)
Author Response
Comments 1: The equations have mixed numbering (some reuse the same numbers multiple times).
Response 1: Thank you for pointing this out. I have reviewed all equation labels and references, ensured that each equation has a unique, sequential number, and updated all in-text cross-references accordingly. All duplication issues have now been resolved.
Comments 2: The test data should be speaker-independent if claiming generalization. This is not clarified.
Response 2: Thank you for this valuable suggestion.Since speaker IDs in the Common Voice Cantonese subset are not publicly available, I were unable to enforce a strictly speaker-independent split. Instead, we partitioned the data into training, development, and test sets according to the official 70%/10%/20% proportions, which may allow speaker overlap between partitions. I have explicitly stated this limitation in the “Experimental Setup and Evaluation Metrics” section.
Comments 3: “wav2vec 2.0” vs “Wav2Vec 2.0” (Inconsistent capitalization)
Response 3: Thank you for catching this inconsistency. I have now standardized every occurrence of the model name to “wav2vec 2.0” (all lowercase) throughout the manuscript—including section titles, table and figure captions, and in-text mentions—to ensure a uniform style.
Comments 4:Uses mixed % and percent (unify style)
Response 4: Thank you for pointing this out. I have replaced all instances of the word “percent” with the “%” symbol for consistency throughout the manuscript. The term “±0.4 percentage points” in the confidence interval is retained because “percentage points” denotes the absolute difference between two percentage values, which is conceptually distinct from a simple “%” notation.All other occurrences have been converted to the “%” symbol.
Comments 5: Comments on the Quality of English Language
Response 5: Thank you for highlighting these language issues. I have corrected “Phoneme Error Rat” to “Phoneme Error Rate,” changed “coarse-grained data-augmentation strategies” to “coarse-grained data augmentation strategies,” and revised “This paper introduce Semantics-Sensitive …” to “This paper introduces ….” All similar instances have been updated accordingly.